# Hilbert Distillation for Cross-Dimensionality Networks

**Dian Qin**[1][*]  **Haishuai Wang**[1][*][†]  **Zhe Liu**[1]  **Hongjia Xu**[1]  **Sheng Zhou**[1,2]  **Jiajun Bu**[1][†]

[1]Zhejiang Provincial Key Laboratory of Service Robot,
College of Computer Science, Zhejiang University, Hangzhou, China
[2]School of Software Technology, Zhejiang University, Ningbo, China
{qindian, haishuai.wang, zheliu, xu_hj, zhousheng_zju, bjj}@zju.edu.cn

## Abstract

3D convolutional neural networks have revealed superior performance in processing volumetric data such as video and medical imaging. However, the competitive performance by leveraging 3D networks results in huge computational costs, which are far beyond that of 2D networks. In this paper, we propose a novel Hilbert curve-based cross-dimensionality distillation approach that facilitates the knowledge of 3D networks to improve the performance of 2D networks. The proposed Hilbert Distillation (HD) method preserves the structural information via the Hilbert curve, which maps high-dimensional (>=2) representations to one-dimensional continuous space-filling curves. Since the distilled 2D networks are supervised by the curves converted from dimensionally heterogeneous 3D features, the 2D networks are given an informative view in terms of learning structural information embedded in well-trained high-dimensional representations. We further propose a Variable-length Hilbert Distillation (VHD) method to dynamically shorten the walking stride of the Hilbert curve in activation feature areas and lengthen the stride in context feature areas, forcing the 2D networks to pay more attention to learning from activation features. The proposed algorithm outperforms the current state-of-the-art distillation techniques adapted to cross-dimensionality distillation on two classification tasks. Moreover, the distilled 2D networks by the proposed method achieve competitive performance with the original 3D networks, indicating the lightweight distilled 2D networks could potentially be the substitution of cumbersome 3D networks in the real-world scenario.

## 1 Introduction

Knowledge distillation aims to transfer knowledge from cumbersome models to lightweight models. The vanilla knowledge distillation [11] collects the logits that contain the cognizance of wrong classes in the cumbersome model. Forcing the lightweight model to mimic the logits can obviously strengthen its parsing ability. Recent efforts further devote to improve the distillation effectiveness by adopting intermediate representations [9, 18, 32, 35, 36] and the relation knowledge [22, 23, 25, 33] of samples as new distillation mediums. In some real-world scenarios (e.g., video analysis and medical imaging processing), 3D models typically present overwhelming performance compared with common 2D models, yet the 3D models suffer from huge computational costs. One way to mitigate this problem is by leveraging knowledge distillation to encapsulate the parsing ability of 3D models into 2D models. However, the cross-dimensionality distillation problem, especially 3D-to-2D distillation, is largely unaddressed in the literature.

Most existing methods that utilize intermediate features for distillation are designed by employing a metric function (e.g., mean square error (MSE) and Kullback-Leibler (KL) divergence) between

---

[*]Equal contribution.
[†]Corresponding authors.

36th Conference on Neural Information Processing Systems (NeurIPS 2022).

the pair of feature maps extracted from teacher and student networks. Unfortunately, they are not applicable in cross-dimensionality distillation tasks due to the inconsistent dimensionality. To get rid of the constraint, several dimensionality reduction methods such as pooling and convolution that can completely condense the extra dimension maybe feasible by converting the 3D feature maps (with the size of $D \times W \times H$) to reduced 2D-from-3D representations (with the size of $1 \times W \times H$). However, the reduction tends to dropout the high-dimensional structural information. As a result, they have limited applicability to deal with volumetric data in the case of lacking the structural information.

In this paper, we propose a new method, namely *Hilbert Distillation (HD)*, to explicitly distill structural information embedded in intermediate features of 3D models. Our approach preserves both 3D and 2D features and maps them into one-dimensional representations based on the intrinsic rules of Hilbert curve [10] in the original feature space. The distilled models with Hilbert curve retain the relative position information in the high-dimensional space, i.e., the mapped data points that are close to each other in the 1D space are also adjacent in the original high-dimensional space, thereby the structural information is highly preserved via the proposed HD method. Hence, the distilled lightweight 2D models obtain significant improvements by learning the converted one-dimensional representations from the 3D feature maps. Furthermore, we propose *Variable-length Hilbert Distillation (VHD)* to more efficiently transfer the structural knowledge by dynamically shortening the walking stride when constructing Hilbert curves in activation areas of feature maps [37]. To address the optimization problem in the dynamic process, we also present an approximation approach by calculating self-adaptive weights when adopting mapping functions of the Hilbert curve. We conduct extensive cross-dimensionality distillation experiments in activity and medical image classification domains, where 3D models are typically applied. The proposed method further narrows the gap between student and teacher networks comparing with the current state-of-the-art in the distillation benchmark. More importantly, the distilled 2D models by our method achieve competitive performance with the 3D models, suggesting that the structural information learned by the proposed method helps promote the generalizability of the 2D models. Hence, the distilled lightweight models could potentially be the substitution of cumbersome models for analyzing the volumetric data.

## 2 Related Work

**Knowledge Distillation**. The vanilla knowledge distillation [11] focuses on transferring knowledge embedded in the logits, i.e., the model outputs scores before applying Softmax. It also provides the idea that coarse information would be beneficial to the distillation, and proposes the Softmax with temperature should be adopted before the calculation of distillation loss. Succeeding efforts [9, 16, 18, 26, 32, 35, 36] try to utilize intermediate representations further. For example, the Attention Transfer [36] calculates attention maps by accumulating values in feature maps along with the channel dimension. The attention maps set out the most informative features for the student model and tackle the problem of different channel sizes between participated models. Some other works [22, 23, 25, 33] try to extract the relation knowledge among samples to enable the distillation between models. For instance, SP [33] calculates the relationships between intermediate feature maps of different inputs. Conducting the distillation between student and teacher relationship maps results in decent improvements to the student model. Recent works tend to explore deeper knowledge and transfer the knowledge in more complicated ways. Ji et al. [14] propose to use feature maps of all layers through the integration with self-adaptive weights learned by attention mechanism. HKD method [38] designs a graph-based scheme to distill holistic knowledge, thus the student model is able to learn individual knowledge and relational knowledge simultaneously.

**Dimensionality Reduction**. Most of the mentioned methods are not applicable for cross-dimensionality distillation networks due to the inconsistent dimension. A straightforward way to cope with this issue is to adopt dimensionality reduction methods. For example, PCA [12] and LDA [6] are commonly used to obtain low-dimensional representation by projecting data points to a few principal components. Some non-linear dimensionality reduction methods such as ISOMAP [31] and LLE [3] are available for more complicated sample distributions. Recent efforts about transformer [4, 27] provide valuable ideas for learning position coding for complex representations such as continuous large-scale feature maps. The reduction can also be performed by following the learned position coding. However, these methods either lose the important structural information for the cross-dimensionality distillation to a certain extent or divide the original high-dimensional space into much smaller space where the problem of dimensional inconsistency still exists. This work refers to

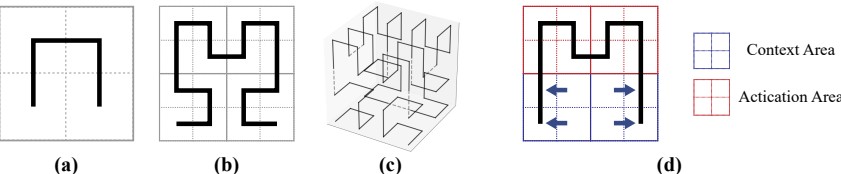

Figure 1: Examples of the Hilbert curve (a-c) and the proposed Variable-length Hilbert curve (d).

a continuous space-filling dimensionality reduction approach, namely Hilbert curve, to address the above issues. Details are described in the next section.

**Cross-dimensionality Distillation**. Recently, cross-dimensionality distillation has also been studied in [2, 13, 19] that have considered adopting 3D models as the distillation participants. However, their methods still follow the 2D-to-2D distillation by applying customized tricks in order to transfer 3D representations into the 2D form, resulting the 2D models are improved only marginally after performing the severely diluted knowledge. To this end, we propose a new cross-dimensionality distillation method by explicitly extracting and transferring the structural knowledge embedded in high-dimensional feature maps.

## 3 Method

In this section, we first introduce the Hilbert Distillation for cross-dimensionality scenarios based on the Hilbert curve. Then we elaborate the proposed Variable-length Hilbert Distillation by dynamically adjusting the stride of Hilbert curve deduction. Since encoding and optimizing the dynamic progress is challenging, we further present an approximation solution by mapping the conventional Hilbert curve into Variable-length Hilbert curve using self-adaptive weights.

### 3.1 Hilbert Curve

The Hilbert curve [10] is a continuous space-filling curve in the Euclidean space, which provides a mapping between 1D and high dimension space while preserves space structure fairly well. The principle behind the Hilbert curve is that the mapped data points which are close to each other in 1D space are also close to each other in the original high dimension space.

The construction [1, 21] of the Hilbert curve can be different. In this paper, we adopt Lindenmayer-System [17] (L-System) to describe the construction that produces walking guides recursively. The system consists of 4 components: 1) two *Variables*, i.e., $A$ and $B$; 2) three *Constants*, i.e., $\triangleright$ denotes move forward, $\oplus$ denotes turn left $90°$, and $\ominus$ denotes turn right $90°$; 3) the *Axiom A*, i.e., the starting point of recursion; 4) two *Production Rules*:

$$Rule\ 1 : A \rightarrow \oplus B \triangleright \ominus A \triangleright A \ominus \triangleright B \oplus \tag{1}$$

$$Rule\ 2 : B \rightarrow \ominus A \triangleright \oplus B \triangleright B \oplus \triangleright A \ominus \tag{2}$$

We denote that order $p$ controls the times of recursion as well as the scale of the curve. When $p = 1$, we can generate the walking guides as $\oplus \triangleright \ominus \triangleright \ominus \triangleright \oplus$ (note that the *Variables A* and *B* are only the placeholder for embedding the rules) based on the *Production Rule* 1 (Eq. 1) from the starting point *Axiom A*. Thus, the simplest Hilbert curve for a $2 \times 2$ square space illustrated in Fig. 1(a) can be constructed by looking right at the start. If $p = 2$, the *Production Rules* 2112 are embedded into $BAAB$ allocated in *Production Rule* 1 separately. Then we can get the walking guides as $\oplus \ominus \triangleright \oplus \triangleright \oplus \triangleright \ominus \triangleright \ominus \oplus \triangleright \ominus \triangleright \ominus \triangleright \oplus \triangleright \oplus \triangleright \ominus \triangleright \ominus \triangleright \oplus \ominus \triangleright \ominus \triangleright \oplus \triangleright \oplus \triangleright \ominus \oplus$. It is important to note that the contiguous *Constants* $\oplus$ and $\ominus$ will be cancelled out and removed from the guides. Hence, the final walking guides will be $\triangleright \oplus \triangleright \oplus \triangleright \ominus \triangleright \triangleright \ominus \triangleright \ominus \triangleright \oplus \triangleright \oplus \triangleright \ominus \triangleright \ominus \triangleright \triangleright \ominus \triangleright \oplus \triangleright \oplus \triangleright \ominus \triangleright$. Fig. 1(b) illustrates the Hilbert curve of order $p = 2$ for a larger square space based on the guides. Similarly, the Hilbert curve for higher order can be easily constructed via this method. It can also be extended and applied in higher dimensional space [1] as shown in Fig. 1(c).

## 3.2 Hilbert Distillation

Since the above mentioned curve is able to well preserve the structural information of its original space, we facilitate the curve to distill knowledge from dimensionally heterogeneous features. Specifically, we aim to supervise the 2D student models by efficiently learning the knowledge that they have never seen in feature maps extracted from the 3D teacher models.

Fig. 2(a) illustrates the pipeline of the proposed Hilbert Distillation framework. Let $(D_t, W_t, H_t)$ and $(W_s, H_s)$ denote the size of the 3D feature map $\eta_{3d}$ and 2D feature map $\eta_{2d}$ respectively. We first leverage the mapping function of Hilbert curve $\mathcal{H}_{n,p}$ (refer to Algorithm 1) to rearrange and stretch the feature maps. Let parameter $n$ denotes the dimension (we use 2 or 3 in this paper, but theoretically it could be larger), and $p$ denotes the order that controls the curve length as described in Sec.3.1. We define the following rule to determine the value of $p$ using the ceiling function $\lceil \cdot \rceil$ :

$$p_{n=3} = \lceil \log_2 \max(D_t, W_t, H_t) \rceil \tag{3}$$
$$p_{n=2} = \lceil \log_2 \max(W_s, H_s) \rceil \tag{4}$$

After determining the values of $n$ and $p$, we adopt the Lindenmayer-System described in Sec 3.1 to generate the Hilbert curve with desired scale, and Algorithm 1 helps deduce the mapping function $\mathcal{H}_{n=2,p}(i,j) = v$ that gives the pixel-level surjective mapping rule from spatial feature map $\eta_{2d}$ to a one-dimensional representation $\hat{\eta}_{2d}$. Although our algorithm describes the case of $n = 2$, it can synergistically be expanded to $n = 3$ to get the 3D mapping function $\mathcal{H}_{n=3,p}(i,j,k) = v$ with the construction of Hilbert curve for 3D space. The output value $v$ is the new position index of the original pixel-level feature in the one-dimensional representation with the length $2^{n \times p}$. Thus, the one-dimensional representations $\hat{\eta}_{3d}, \hat{\eta}_{2d}$ of the feature maps $\eta_{3d}, \eta_{2d}$ can be calculated by

$$\hat{\eta}_{3d}(v) = \eta_{3d}(i,j,k) \quad \text{where} \quad \mathcal{H}_{n=3,p}(i,j,k) = v \tag{5}$$
$$\hat{\eta}_{2d}(v) = \eta_{2d}(i,j) \quad \text{where} \quad \mathcal{H}_{n=2,p}(i,j) = v \tag{6}$$

At this point, we can define the Hilbert Distillation between cross-dimensionality features by calculating the loss between the converted one-dimensional representations as

$$\mathcal{L}_{hd} = \left\| \frac{\mathcal{R}(\hat{\eta}_{3d})}{||\mathcal{R}(\hat{\eta}_{3d})||_2} - \frac{\mathcal{R}(\hat{\eta}_{2d})}{||\mathcal{R}(\hat{\eta}_{2d})||_2} \right\|_1 \tag{7}$$

where $\mathcal{R}(\cdot)$ is the nearest rescaling function that scales the length of $\hat{\eta}_{3d}$ from $2^{3 \times p}$ to $2^{2 \times p}$ before the calculation, and $||\cdot||_2$ and $||\cdot||_1$ are the L2 and L1 normalization. The division between representations and its L2 norm is intended to eliminate the distribution difference between features from different modality. As a knowledge distillation method, the distillation process is commonly performed along with the training process. The final loss function for the end-to-end training scheme is given by

$$\mathcal{L} = \mathcal{L}_{CE} + \alpha \mathcal{L}_{hd} \tag{8}$$

where $\mathcal{L}_{CE}$ is the cross entropy loss corresponding to the original task of the student network, and the hyperparameter $\alpha$ is a balance weight between training from distillation and hard labels.

As claimed in the interpretability [15, 28, 37] and attention [36] research areas, only partial feature maps are activated and crucial for the final task. This conclusion implies the reason why our method works. In other words, the superiority of the proposed distillation method is that the 2D models can directly learn the activation features that preserve the 3D structural information as much as possible. As illustrated in Fig. 2(a), the activated "dog features" in 3D feature maps (leftmost) are still gathered together after the mapping, as shown in the black area of the bar. Theoretically, our method could be extended to conduct distillation between networks of any dimensionality since the Hilbert curve-based mapping approach can be applied in any Euclidean space.

In addition, we also provide several tricks to address the following possible minor obstacles when applying our method in real-world scenarios:

**a. What if the activation features in the converted one-dimensional representations of 2D and 3D models are not relatively aligned.** Fig. 2 demonstrates an ideal case that both dogs are located at the left bottom so that the relative positions of the activation features are well aligned. In reality, in the medical imaging task that 3D models are commonly applied, the spatial distribution of human organs are always fixed regardless of the data modality. The activation features from 2D and 3D

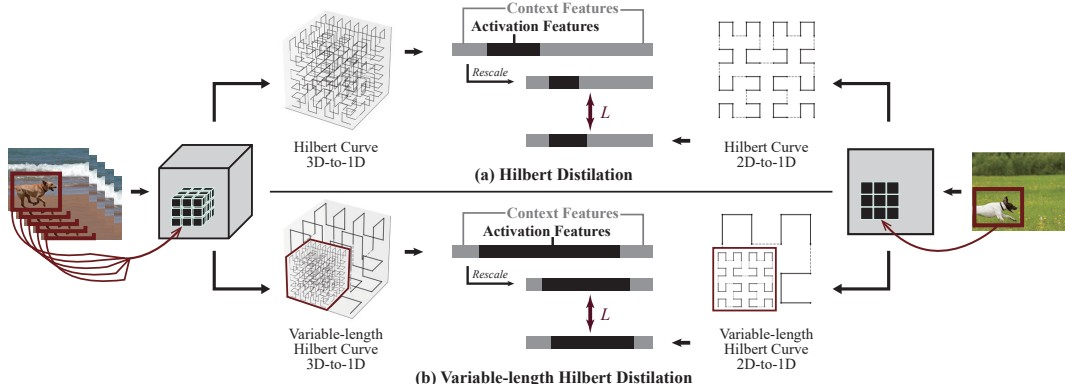

Figure 2: The pipeline of the proposed Hilbert Distillation and Variable-length Hilbert Distillation between cross-dimensionality feature maps.

---

**Algorithm 1:** The Mapping Function of Hilbert Curve $\mathcal{H}_{n=2,p}$

---

1  Initialize $i = 0, j = 0, v = 0$
2  Save the map $\mathcal{H}_{n=2,p}(i, j) = v$
3  **while** *Algorithm 1 is running* **do**
4      // Algorithm 1 starts with looking right
5      **if** $\triangleright$ *is activated* **then**
6          **if** *look up / down* **then**
7              $i = i + 1$   /   $i = i - 1$
8          **else if** *look right / left* **then**
9              $j = j + 1$   /   $j = j - 1$
10         $v = v + 1$
11         Save the map $\mathcal{H}_{n=2,p}(i, j) = v$
12     **else**
13         Waiting for $\triangleright$
14 Output $\mathcal{H}_{n=2,p}$

---

feature maps are well aligned in the final stage of our method. For another task using 3D models, such as video recognition, a simple length-preserving fully connection layer (FC) for any one-dimensional representation before the distillation loss calculation is able to address this issue.

**b. What if the feature maps do not exactly fulfill a square or cube space**. As the vanilla Hilbert curve is only applicable in the space with a side length equals a power of 2, we always construct Hilbert curve for a minimum space that can hold the feature maps according to the side length which can be calculated by Eqs. 3 and 4, e.g., constructing curve for $8 \times 8 \times 8$ space according to a $5 \times 7 \times 7$ feature map. The mapping function $\mathcal{H}$ only considers the areas where contains features.

### 3.3 Variable-length Hilbert Distillation

The scale of activation features is relatively small compared with that of context features in most cases, as the ratio of black and gray parts illustrated in Fig. 2(a). The proportion of activation features in 3D feature maps may even lower than in 2D feature maps. To make the distillation more focus on the activation areas where the features are significantly related to semantics, we further propose an improvement for the construction process of Hilbert curve called Variable-length Hilbert Distillation.

Specifically, we dynamically change the walking stride when constructing the Hilbert curve. As demonstrated in Fig. 2(b), we lengthen the walking stride in areas that consist of context features to draw sparser curves, and shorten the stride in areas that consist of activation features to draw denser curves. Those skipped context features will be represented by the nearest valid features located on the curves. Thus, the proportion of activation features in the calculation of the final distillation loss is

greatly amplified, as depicted in the figure. To this end, we propose to alter the *Production Rules* (Eqs. 1 and 2) in the described L-system into the formation as

$$A \rightarrow \oplus[\phi(B)][\gamma_\phi \rhd] \ominus [\phi(A)][\gamma_\phi \rhd][\phi(A)] \ominus [\gamma_\phi \rhd][\phi(B)]\oplus \tag{9}$$

$$B \rightarrow \ominus[\phi(A)][\gamma_\phi \rhd] \oplus [\phi(B)][\gamma_\phi \rhd][\phi(B)] \oplus [\gamma_\phi \rhd][\phi(A)]\ominus \tag{10}$$

where $\phi(A|B)$ is the decision function that decides whether to perform recursion, which is given by

$$\phi(A|B) = \begin{cases} A|B, & \text{if upcoming walking routes that follow the Hilbert curve contain an activation feature,} \\ Null, & \text{otherwise.} \end{cases} \tag{11}$$

In order to figure out whether there is an activation feature, we define the Activation Mapping (AM) for a given convolutional layer $l$. We first compute the activation weight $\gamma$ for a single feature map in the layer $l$. For 2D and 3D models, the computations are separately defined as

$$\gamma_{3d}^n = \frac{1}{z_{3d} \times c} \sum_{z_{3d}} \sum_c \frac{\partial y^c}{\partial \eta_{3d}^n(i,j,k)}, \quad \gamma_{2d}^m = \frac{1}{z_{2d} \times c} \sum_{z_{2d}} \sum_c \frac{\partial y^c}{\partial \eta_{2d}^m(i,j)} \tag{12}$$

where $m$ and $n$ denote the numbers of feature maps, $z_{3d}$ and $z_{2d}$ are the numbers of features in the 3D and 2D feature maps, $\frac{\partial y^c}{\partial \eta}$ is the gradient of the output score $y^c$ of class $c$ with respect to feature maps $\eta$. Different from the Class Activation Mapping (CAM) proposed by [28], we compute the gradients for all the objective classes and accumulate the results because the knowledge distillation method requires preserving informative knowledge as much as possible. The activation features for wrong classes may also be helpful to the distilled model. To this end, we calculate the AM by weighted sum of the feature maps, which is given by

$$AM_{3d} = \sum_n \gamma_{3d}^n \eta_{3d}^n, \quad AM_{2d} = \sum_m \gamma_{2d}^m \eta_{2d}^m \tag{13}$$

The AM generally presents the importance of features in corresponding positions, and the size of AM is the same as the size of the feature map in the layer $l$. Moreover, we get the activation if the feature's *Sigmoid* value in AM at the corresponding position is larger than a threshold $\theta$. The threshold is user defined based on the required degree of activation.

Fig. 1.(d) illustrates a case when order $p = 2$. For the first round of rule generation with Eq. 9, the inferred walking guides can be $\oplus[\gamma_\phi \rhd] \ominus A[\gamma_\phi \rhd]A \ominus [\gamma_\phi \rhd]\oplus$. Only the two areas marked "*A*" where contain the activation feature are preserved after the execution of $\phi$. The additional factor $\gamma_\phi$ in the guides controls the multiple compensation for the walking steps, which doubles the execution number of the following $\rhd$ whenever a neighboring $\phi$ results in *Null*. Hence, the actual walking guides of the first round are $\oplus \rhd \rhd \ominus A \rhd A \ominus \rhd \rhd \oplus$. In the next round of generation, we can get the final walking guides as $\oplus \rhd \rhd \rhd \ominus \rhd \ominus \rhd \oplus \rhd \oplus \rhd \ominus \rhd \ominus \rhd \rhd \rhd \oplus$ by replacing $A$ with Eq. 9 recursively. This idea is the desired Variable-length Hilbert curve depicted in Fig. 1.(d). The Variable-length Hilbert Distillation (VHD) can be hereby designed by following the framework of HD and substituting the curve construction process, as the illustrated pipeline in Fig. 2(b).

However, unlike the vanilla Hilbert curve that offers the hard mapping for a space of fixed scale, the Variable-length Hilbert curve is dynamically changed in terms of different activation maps from different inputs, however, it is hardly be performed in forward propagation and optimized in backpropagation. In this paper, we propose an approximation approach to achieve the variable-length by calculating self-adaptive weights when adopting the mapping function $\mathcal{H}$. We rewrite the original mapping process in Eqs. 5 and 6 using the product of the features and the values in AM, as follows,

$$\bar{\eta}_{3d}(v) = \eta_{3d}(i,j,k) \cdot AM_{3d}(i,j,k) \quad \text{where} \quad \mathcal{H}_{n=3,p}(i,j,k) = v \tag{14}$$

$$\bar{\eta}_{2d}(v) = \eta_{2d}(i,j) \cdot AM_{2d}(i,j) \quad \text{where} \quad \mathcal{H}_{n=2,p}(i,j) = v \tag{15}$$

Finally, the optimizable distillation loss function of the proposed approximation scheme is given as

$$\mathcal{L}_{vhd} = \left\| \frac{\mathcal{R}(\bar{\eta}_{3d})}{||\mathcal{R}(\bar{\eta}_{3d})||_2} - \frac{\mathcal{R}(\bar{\eta}_{2d})}{||\mathcal{R}(\bar{\eta}_{2d})||_2} \right\|_1 \tag{16}$$

Similar with HD, we define the end-to-end training of VHD by replacing $\mathcal{L}_{hd}$ with $\mathcal{L}_{vhd}$ in Eq.8. Note that we only keep the distilled 2D model during the inference phase.

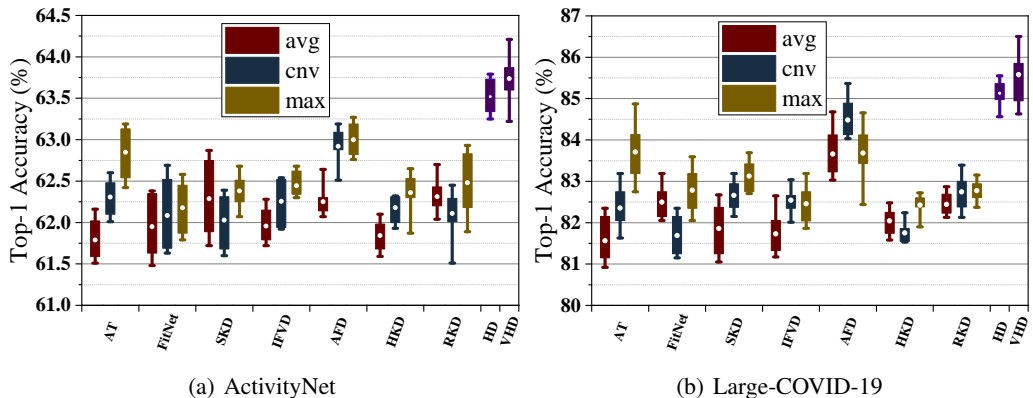

| (a) ActivityNet | (b) Large-COVID-19 |

Figure 3: The performance of distillation methods using feature maps on the ActivityNet (graph a) and Large-COVID-19 (graph b) datasets. The white point centered in the box is the mean value.

# 4 Experiments

## 4.1 Datasets

3D models present overwhelming performance in some specific tasks, such as video recognition and medical imaging processing. Therefore, we demonstrate the effectiveness of the proposed cross-dimensionality distillation method on two datasets: ActivityNet [5] dataset for activity classification and Large-COVID-19 [20] dataset for medical imaging classification.

**ActivityNet**, which is a benchmark video dataset for untrimmed human activity classification. We adopt the ActivityNet-100 version that contains 100 human activity classes. We randomly split the 7000 videos into 2:1 for training and testing.

**Large-COVID-19**, which is a lung Computed Tomography (CT) dataset of COVID-19 for Community Acquired Pneumonia (CAP) classification. This dataset contains 7593 COVID-19 images, 2618 CAP images, and 6893 normal images. Most images can make up the 3D CT scan by combining their neighboring images. We randomly select 20% images as the test set.

## 4.2 Experiment Setting

**Distillation Setup**. Different from existing distillation works that consider the same dimensionality in the teacher and student models, we adopt ResNet-50 [8] and VGG16 [30] as the 2D student models and well-trained 3D ResNet-50 as the 3D teacher model. In the training process, we feed the randomly selected images into the 2D models, and the volumetric data that consists of 16 consecutive frames into the 3D model. The selected pair of 2D and 3D inputs are restricted in the same category. Since the category of the entire video (activity) is hard to be classified using only one frame, the 2D training scheme of video processing commonly adopts the combination of 2D CNN + LSTM [29] or 2D CNN with voting strategies. We follow the former scheme to train the 2D student model by sequentially feeding one of the 16 consecutive frames that are prepared for the 3D teacher model into the 2D student model. The distillation method is applied between 3D CNN and 2D CNN before performing the voting. We train and test models on an NVIDIA GeForce RTX 3090 GPU (24GB).

**Network Architectures**. For the activity classification task, we randomly crop the input frame into $256 \times 256$. The batch size is set to 16. The training process lasts 40 epochs. The hyperparameter $\alpha$ in Eq. 8 is set to $10^3$. For the medical imaging classification task, we resize the input image into $224 \times 224$ horizontally. The batch size for training is set to 32. The number of training epochs is 60. Different from the activity classification task, we set $\alpha$ to 10. The selection and performance of the fluctuation of $\alpha$ are also discussed in our experiments. The models are trained with an initial learning rate 0.01, and the Adam optimization algorithm is employed for both tasks.

Table 1: Top-1 accuracy (%) of the student models on the ActivityNet and Large-COVID-19 with distillation. The **bold** values are the best distillation performance.

| Teacher Student | ActivityNet | | | Large-COVID-19 | | |
|---|---|---|---|---|---|---|
| | 3DResNet-50 ResNet-50 | 3DResNet-50 VGG16 | ARI(%) | 3DResNet-50 ResNet-50 | 3DResNet-50 VGG16 | ARI(%) |
| **Teacher** | 71.34 | 71.34 | \ | 90.15 | 90.15 | \ |
| **Student** | $61.42 \pm 0.18$ | $60.22 \pm 0.09$ | | $79.92 \pm 0.02$ | $77.4 \pm 0.12$ | |
| KD [11] | $62.30 \pm 0.29$ | $61.45 \pm 0.38$ | 184.59 | $82.08 \pm 0.53$ | $82.37 \pm 0.12$ | 110.51 |
| SP [33] | $62.88 \pm 0.36$ | $62.27 \pm 0.51$ | 71.11 | $83.69 \pm 0.32$ | $82.85 \pm 0.46$ | 47.79 |
| PKT [23] | $62.73 \pm 0.48$ | $62.70 \pm 0.29$ | 64.02 | $83.20 \pm 0.26$ | $82.69 \pm 0.23$ | 61.15 |
| RKD [22] | $62.14 \pm 0.67$ | $61.23 \pm 0.46$ | 247.15 | $83.02 \pm 0.45$ | $82.44 \pm 0.91$ | 69.87 |
| CCKD [25] | $62.78 \pm 0.49$ | $61.88 \pm 0.33$ | 98.65 | $83.19 \pm 0.41$ | $82.70 \pm 0.58$ | 61.27 |
| HD | $63.55 \pm 0.28$ | $63.46 \pm 0.49$ | 12.40 | $85.05 \pm 0.58$ | $84.63 \pm 0.38$ | 9.99 |
| VHD | $\mathbf{63.71 \pm 0.63}$ | $\mathbf{64.02 \pm 0.85}$ | 0 | $\mathbf{85.55 \pm 0.72}$ | $\mathbf{85.37 \pm 0.86}$ | 0 |

## 4.3 Results and Analysis

To demonstrate the performance and stability of the commonly used distillation methods, we separate the experiments into two parts based on their applicability in the cross-dimensionality distillation problem. We first present the performance of the distillation methods using feature maps, i.e., AFD [14], SKD [18], FitNet [26], IFVD [34], HKD [24], RKD [7], and AT [36]. Since they are not to be applied directly due to the dimensionality difference problem, we employ three alignment functions, i.e., average pooling ("avg" for short in the figure), max pooling ("max" for short in the figure), and convolution with kernel size $D \times 1 \times 1$ ("cnv" for short in the figure), to align the 3D feature maps into 2D representations by compacting the values along the dimension $D$. Fig. 3 shows the Top-1 accuracy (%) of baselines and the proposed methods for the activity and medical imaging classification tasks. The range values in the boxes are collected from the last ten epochs of all folds. The distillation performance is more stable if the box is shorter. The proposed methods HD and VHD outperform all the baselines in terms of all the tasks. Compared with existing methods, the accuracy of the proposed HD method improves about 2% on the Large-COVID-19 data and 1.5% on the ActivityNet using both homogeneous architecture (3DResNet to ResNet) and heterogeneous architecture (3DResNet to VGG). The better performance of the proposed VHD method implies that VHD is able to transfer the activation knowledge more efficiently. We can also conclude that the max pooling and convolution alignment functions should be prioritized when the conventional distillation approaches are not able to deal with the dimensionality inconsistency in the cross-dimensionality distillation problem.

The goal of many distillation methods (i.e., RKD [22], PKT [23], CCKD [25], and SP [33]) is to excavate the relation knowledge among the data samples. Theoretically, they can be directly applied in cross-dimensionality problems because the calculation between the feature maps of the data can be performed inside the model. The scale of calculated relation representations for distillation is only related to the number of samples. The premise to be effective of those methods is that the inputs of the distillation models are consistent, and the student models can learn relational knowledge from teacher models. However, this condition does not hold in the cross-dimensionality scenario. Table 1 provides the performance of commonly adopted methods. We can observe that the proposed HD/VHD methods achieve a remarkable improvement compared with the benchmark methods. The main reason why the baselines do not perform well is that the data prepared for 3D and 2D models are usually different. Note that the vanilla KD [11] is always feasible if distillation is applied in classification tasks because the number of logits is only related to the number of classes. The performance of the vanilla KD can also be regarded as a benchmark of distillation methods for cross-dimensionality problems.

As shown in Table 1, the fluctuation range of VHD is greater than HD. The main reason might be the Variable-length Hilbert curve is not directly obtained in this work. We approximately achieve the curve by the products of the corresponding values in Activation Mapping, which is the same as the self-adaptive weights used for the vanilla Hilbert curve. Although the values deduced by the gradients of output scores do not perfectly indicate the activation features, VHD always outperforms others in all experiments. As demonstrated in Fig. 3(b), only VHD can reach over 86% Top-1 accuracy. To show the advantages of our method more clearly, we calculate the Average Relative Improvement

(ARI) by

$$\text{ARI} = \frac{1}{M} \sum_{i=1}^{M} \frac{\text{Acc}_{\text{VHD}}^{i} - \text{Acc}_{\text{BKD}}^{i}}{\text{Acc}_{\text{BKD}}^{i} - \text{Acc}_{\text{STU}}^{i}} \times 100\% \tag{17}$$

where $M$ is the number of different architecture combinations, and BKD and STU refer to the baseline methods and student network, respectively. As shown in Table 1, ARI presents the magnitude of the improvement of the proposed VHD compared with the method located in that row.

## 4.4 Hyperparameter Tuning

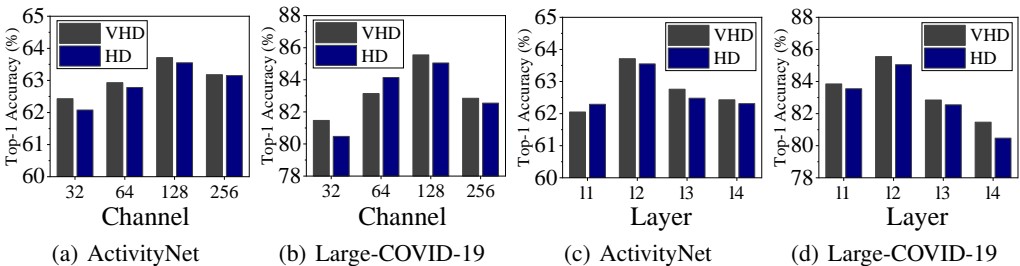

|     |     |     |     |
| --- | --- | --- | --- |
| (a) ActivityNet | (b) Large-COVID-19 | (c) ActivityNet | (d) Large-COVID-19 |

Figure 4: Performance of proposed methods HD and VHD with respect to various hyperparameters. Results are from the ResNet-50 model distilled from a 3D ResNet-50.

In addition to discussing the effectiveness of our methods, we also conduct tuning experiments to demonstrate the influence of the hyperparameters. We follow the setup described above, and adopt a 3D ResNet-50 as the teacher model to train a ResNet-50 as the devised distillation method. Unlike the range values shown above, here we only present the mean Top-1 accuracy (%) of the distilled student networks to compare the fluctuation more clearly.

**Hyperparameters in the loss functions**. As shown in Eq. 8, the hyperparameter $\alpha$ controls the tradeoff of the original task and distillation during training. Fig. 5 illustrates the performance with respect to various $\alpha$ for both two tasks. The value of $\alpha$ could be initialized with a relatively large value, i.e., about $10^3$ for the medical imaging classification and 10 for the activity classification task. Extensive experiments demonstrate that our method can perform well in the cross-dimensionality problems given any value within a reasonable range.

**The number of feature maps for distillation**. Sec. 3 introduces the distillation using a pair of feature maps extracted from 3D and 2D models. In most cases, a given layer can generate the same number of feature maps as the convolutional kernel. The distillation loss in Eq. 8 can also be calculated in batches between the same number of dimensionally heterogeneous feature maps. For architecture homogeneous models such as 3D ResNet and ResNet, the number of feature maps is naturally the same as in the symmetrically positioned layer. For architecture heterogeneous models, we suggest applying extra $1\times1$ convolutional layers to align the channel dimension, i.e., the number of feature maps. Figs. 4(a) and (b) show the performance of our method with respect to various number of feature maps. The highest score in both figures is reached when using 128 feature maps for distillation.

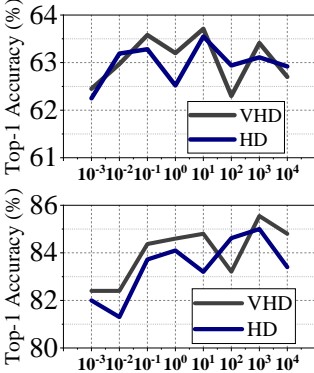

Figure 5: Performance with respect to various $\alpha$ on the ActivityNet (top) and Large-COVID-19 (bottom).

**Layer selection for distillation**. Finally, we show the effectiveness of our methods with respect to various convolutional layers. Figs. 4(c) and (d) illustrate that the distillation on the second layer (resblock) contributes to the most improvement for the student model. It is important to note that the layer distribution (from layer 1 to layer 4) in our experiments is from ResNet, thereby this result is shown for reference only, if a different model architecture is applied. In conclusion, rear layers whose feature

maps are not too small should have priority to be distilled because very small feature maps usually do not have explicit structural knowledge.

## 5 Conclusion

Although a considerable improvement has been obtained using cumbersome models when analyzing volumetric data, the success of cumbersome models comes primarily from the availability of computing power. Our goal is to distill the cumbersome models into lightweight models while preserving the structural information in the high-dimensional space. To this end, we propose a Hilbert curve-based approach, which aims to transfer structural information as much as possible from 3D to 2D models. Furthermore, we design the Variable-length Hilbert Distillation to force the student to focus more on learning the structural knowledge from activation features. Extensive experiments demonstrate the effectiveness of the proposed methods.

## Acknowledgements

This work is supported by the National Natural Science Foundation of China under Grants 61972349, 62202422 and 62106221.

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
