# OpenReview forum: "Hilbert Distillation for Cross-Dimensionality Networks"
_NeurIPS.cc/2022/Conference — NeurIPS 2022 Accept_

### Official Review · Reviewer_ELut · 2022-06-30

**Rating:** 7
**Confidence:** 4
**Soundness:** 3 good
**Presentation:** 3 good
**Contribution:** 2 fair

**Summary:**

In this paper the authors present a cross-dimensional distillation approach. More specifically, they proposed to use the knowledge of 3D networks to improve the performance of 2D networks. TO this end, they propose to present the structural information that exists in the teacher model using Hilbert curves, which can map the high dimensional representations to one dimensional continuous filling curves. The student network is then supervised using this information. To further improve the performance of the proposed method they also propose a variable length variant of the proposed method which can dynamically shorten the walking stride of the Hilbert curve. The authors demonstrate that the proposed method outperforms the current distillation methods that can be used for cross-dimensionality distillation using two datasets.

**Questions:**

Generally I am positive regarding this paper, since it introduces a novel idea on how cross-dimensional distillation can be performed. However, there is clear room for improvements, e.g., compare with regular distillation from larger 2D networks to smaller ones, evaluate the effect of using different layers - different layer matching, discuss the computational complexity, extend the evaluation to other larger datasets.

**Limitations:**

Authors discuss limitations of their work. However, there is no dedicated limitations section.

**Strengths And Weaknesses:**


Positive aspects:

- The idea proposed in this paper is interesting and makes sense. The paper is well written and easy to follow.
- The experimental evaluation clearly demonstrates the improvements obtained using the proposed method. Ablation studies are also provided.

Negative aspects:
- There are no comparisons with regular distillation approaches from larger 2D networks to smaller 2D networks. Only cross dimensional distillation is evaluated - if I have understood the experiments correctly.
- It is not clear how the layer that is used for extracting the features can affect the distillation. How should the layers be matched to each other?
- There is no discussion on the computational complexity of generating the Hilbert-based matching during the training.
- Only two -relatively small - datasets were used for the evaluation. As far as I understand the proposed method can be used for any video dataset.

---

> ### Author Response · Authors · 2022-08-02
> **Response to reviewer Elut -- part 1**
>
> Many thanks for your valuable comments and constructive feedback! We will answer your concerns in the following.
>
> >Q1: There are no comparisons with regular distillation approaches from larger 2D networks to smaller 2D networks. Only cross dimensional distillation is evaluated - if I have understood the experiments correctly.
>
> A1: Thanks for your comment. Since 3D-to-2D distillation problem has been rarely studied, the baselines in our experiments are actually 2D-to-2D distillation approaches. To the best of our knowledge, there is no efficient solution for the cross-dimensionality distillation problem so far. In our experiments, we employ the regular 2D-to-2D distillation approaches to adapt to 3D-to-2D scenarios for comparison by leveraging some extra operations such as alignment functions. When considering dealing with 3D-to-2D distillation, we categorize existing 2D-to-2D distillation methods into three classes, as follows:
>
> 1. Traditional 2D-to-2D intermediate feature maps distillation methods, such as AT, FitNet, SKD, IFVD, and AFD in our experiment (Figure 3). Since they cannot to be applied directly due to the challenge of the dimensionality difference, we employ alignment functions (i.e., average pooling, max pooling, and convolution) to align the 3D feature maps into 2D representations to enable the 3D-to-2D distillation as described in Line # 261 - 266.
> 2. Traditional 2D-to-2D relation knowledge distillation methods, such as PDK, PKT, CCDK, and SP in our experiment (Table 1). They can be directly applied to 3D-to-2D distillation because the calculation can be performed inside the model. We have discussed them in Line # 277- 281.
> 3. Contrastive learning based distillation methods [1, 2]. The reason why they are not included in the experiment is because the student requires the same/augmented samples of the teacher to construct “positive pair”, which is impossible in cross-dimensionality distillation as the input dimension is different.
>
> We agree that more detailed discussion about the adaptability of 2D-to-2D distillation methods on 3D-to-2D distillation problems in our paper is necessary. We have added the discussion in the appendix of the revision. Please refer to Appendix "C. The Adaptability of 2D-to-2D Distillation Methods on 3D-to-2D Distillation Problems" in the revised paper.
>
> >Q2: It is not clear how the layer that is used for extracting the features can affect the distillation. How should the layers be matched to each other?
>
> A2: Thanks for your question. We have provided the effectiveness of our methods with respect to feature maps extracted from different convolutional layers in the ResNet-like architecture in the initial submission. Please refer to Sec.4.4 "Hyperparameter Tuning - Layer selection distillation" (Line # 323 - 331), Figure 4(c), and Figure 4(d).
>
> By leveraging the Hilbert mapping function, we can keep the information of the feature maps globally because the Hilbert curve is a surjective mapping function. Therefore, features of a student layer can receive valid information from any features in the teacher layer. However, choosing the layer pairs in the relatively same location (the former, the middle, or the latter) is recommended to reach the better distillation performance. Some recent works [3] have discussed the utilization of different blocks simultaneously. However, it is not the scope of our paper.
>
> >Q3: There is no discussion on the computational complexity of generating the Hilbert-based matching during the training.
>
> A3: Thanks for you advice. In fact, the computational cost from generating Hilbert Curve to finishing the Hilbert-based mapping function $\mathcal{H}_{n,p}$ is very low. In the revision, we ran the generation processor on a single process of the Intel(R) Xeon(R) Silver 4216 CPU (2.10GHz) and calculated the computational costs on different sizes of feature maps. Results are presented as follows.
>
> | Time Consuming (ms)  |      |      |      |      |      |      |      |       |
> |---------------------|-------|-------|-------|-------|-------|-------|-------|--------|
> | $S$ (side length)      | 2     | 4     | 8     | 16    | 32    | 64    | 128   | 256    |
> | 2D (# points = $S^2$) | 0.034 | 0.046 | 0.090 | 0.204 | 0.458 | 2.505 | 2.319 | 5.166  |
> | 3D (# points = $S^3$) | 0.048 | 0.072 | 0.159 | 0.376 | 0.874 | 3.359 | 4.550 | 10.043 |
>
> What can be observed is that the mapping only costs 10ms even in processing a 3D feature map with the size 256 $\times$ 256 $\times$ 256. In reality, the time complexity of the generation process depends on the implementation of the Hilbert Curve. We adopt the most common approach to generate Hilbert Curve in $O(\log{n})$ time. We agree that detailed discussion about the computational complexity of the generation process is necessary. Thus, we have added the discussion in the appendix of the revision. Please refer to Appendix "B. Computational Costs of Generating Hilbert Mapping Function" in the revised paper.

---

> ### Author Response · Authors · 2022-08-02
> **Response to reviewer Elut -- part 2**
>
> >Q4: Only two -relatively small - datasets were used for the evaluation. As far as I understand the proposed method can be used for any video dataset.
>
> A4: Thanks for your advice. We have continued some extra experiments on Kinetics-400 after the submission. Since they are not included in the original submission, we listed the table in the following:
>
> | Kinetics-400   |                 |                 |
> |----------------|------------------|------------------|
> | Teacher        | 3DResNet-50      | 3DResNet-50      |
> | Student        | ResNet-50        | VGG16            |
> | Teacher        | 74.15            | 74.15            |
> | Student        | 67.20 $\pm$ 0.23 | 65.43 $\pm$ 0.19 |
> | KD             | 68.03 $\pm$ 0.31 | 66.71 $\pm$ 0.46 |
> | SP             | 69.14 $\pm$ 0.48 | 68.29 $\pm$ 0.55 |
> | PKT            | 68.37 $\pm$ 0.29 | 67.35 $\pm$ 0.41 |
> | AT (with avg)  | 68.21 $\pm$ 0.59 | 66.86 $\pm$ 0.30 |
> | AT (with cnv)  | 68.35 $\pm$ 0.44 | 67.14 $\pm$ 0.47 |
> | AT (with max)  | 68.13 $\pm$ 0.32 | 67.19 $\pm$ 0.44 |
> | AFD (with avg) | 68.82 $\pm$ 0.65 | 67.77 $\pm$ 0.58 |
> | AFD (with cnv) | 69.59 $\pm$ 1.02 | 67.90 $\pm$ 0.81 |
> | AFD (with max) | 69.08 $\pm$ 0.64 | 68.48 $\pm$ 0.59 |
> | HD (ours)      | 70.28 $\pm$ 0.36 | 69.82 $\pm$ 0.42 |
> | VHD (ours)     | 70.91 $\pm$ 0.85 | 70.40 $\pm$ 0.73 |
>
> The results demonstrated that our method still outperforms the best performance holder AFD and SP in the current completed experiments. We have added the results in the appendix of the revision. Please refer to Appendix "D. More Benchmarks" in the revised paper. We will also present the experiments on Kinetics-400 of the same scale as the existing ActivityNets experiment in the final version.
>
>
> Reference
>
> [1] Contrastive representation distillation
>
> [2] Contrastive multiview coding
>
> [3] Cross-Layer Distillation with Semantic Calibration

---

### Official Review · Reviewer_qHTA · 2022-07-08

**Rating:** 7
**Confidence:** 3
**Soundness:** 1 poor
**Presentation:** 1 poor
**Contribution:** 3 good

**Summary:**

The paper propose a Hebert curve-based approach to perform distillation of the large models into lightweight models. In specific, the paper proposes Variable-length Hilbert curve that if flexible for different activation maps from different inputs. This is different than vanilla Hilbert curve that offers hard mapping for a space of fixed scale. The efficacy of the methods is demonstrated in two separate benchmarking datasets where the proposed method achieve improved performance over existing methods.


**Questions:**

Check weakness section above.

**Limitations:**

Check weakness section above.

**Strengths And Weaknesses:**

Strength:
- The paper proposes a novel approach based on Hibert curve to perform distillation to facilitate the knowledge of 3D networks to improve the performance of 2D networks. The idea is interesting as it doesn't require computational scale as the existing methods.
- The proposed Hilbert Distillation method is extended as Variable-length Hilbert Distillation (VHD) to provide more flexibilityin activation feature map and stride length.
- The experimental evaluation demonstrate improved performance.

Weakness:
- The major weakness is the poor writing and presentation of the paper. This not only made reading paper difficult but also made it impossible to understand the results and outcomes of the proposed method. Some highlights of this include:
a. Poor and unclear figures: Fig 3, Fig 4 and Fig 5 is missing legends, X- and Y-axis. It is impossible to understand what's happening.
b. Grammatical issues in the writing.
Overall, the submission quality is far below the quality of NeurIPS submission. I strongly encourage authors to carefully proofread their paper before submission.

- The experimental setup also is confusing.
a. Is ActivityNet also splitted randomly? The wordings in the description of two datasets raises the question.
b. Also why different ratio of train-test split is done for two dataset?
c. How are hyperparameter selected? e.g., alpha, epochs number, batch-size, etc.


######### update ##########
There was technical issue on my end which impacted my initial reviews. After resolving this technical issue, I was able to understand the paper more clearly. With this understanding, I am increasing my score to 7.

---

> ### Author Response · Authors · 2022-08-02
> **Response to reviewer qHTA**
>
> Thanks for your valuable time. We have read your review carefully, and, unfortunately, found that there maybe existing some factual errors in this comment. We will answer your concerns and explain the potential misunderstanding in the following.
>
> >Q1: The major weakness is the poor writing and presentation of the paper. This not only made reading paper difficult but also made it impossible to understand the results and outcomes of the proposed method. Some highlights of this include: a. Poor and unclear figures: Fig 3, Fig 4 and Fig 5 is missing legends, X- and Y-axis. It is impossible to understand what's happening. b. Grammatical issues in the writing. Overall, the submission quality is far below the quality of NeurIPS submission. I strongly encourage authors to carefully proofread their paper before submission.
>
> A1: Thanks for your comment, but we hope you can double-check this problem because we believe there maybe existing factual errors in this comment. First, although there are some grammar errors and typos, all other reviewers comment our presentation is good/excellent. One of reviewers points out that this paper is well written and easy to follow. We will definitely carefully proofread our paper in the final revision. Second, we argue that your description "Fig 3, Fig 4 and Fig 5 is missing legends, X- and Y-axis" is a factual error. All the three figures have the legends, X- and Y-axis in our original submission. Although Fig. 5 does not have the explicit X-axis notation, we have described in the caption that this figure shows the straightforward experiments about the influence of alpha (in other words,  it is the values of alpha on the X-asis).
>
> >Q2: The experimental setup also is confusing. Q2. a. Is ActivityNet also splitted randomly? The wordings in the description of two datasets raises the question. Q2. b. Also why different ratio of train-test split is done for two dataset? Q2. c. How are hyperparameter selected? e.g., alpha, epochs number, batch-size, etc.
>
> A2. (a): Yes, The ActivityNet was also split randomly. We missed the word "randomly" and have added it in the revised paper.
>
> A2. (b): We followed the previous works (e.g., [1]) that also use the ActivityNet dataset to split training and testing sets. The adopted split of train/test sets has been widely used in ActivityNet Challenge. For the Large-COVID-19 dataset, we allocated more images to training sets to achieve sufficient training due to the relatively low scale of medical imaging datasets.
>
> A3. (c): In fact, all the hyperparameter selection and settings have been provided in “Sec. 4.2 Experiment Setting - Network Architectures”. This section has detailed described how to select alpha, epochs number, batch size, etc. Please refer to Line # 251 - 257 if you missed it, and please read this section in our paper.
>
> We hope the above responses can address your concerns and misunderstandings (if exists). We look forward to furthering discussions with you.
>
> Reference
>
> [1] CBR-Net: Cascade Boundary Refinement Network for Action Detection: Submission to ActivityNet Challenge 2020 (Task 1).

---

> > ### Comment · Reviewer_qHTA · 2022-08-07
> > **Response to authors**
> >
> > Dear authors,
> >
> > Please accept my apologies. My "Mendeley Desktop" app seems to remove the axis information, legend information, etc., from all the figures you provided in the paper. And surprisingly, this has happened only to your paper. This issue directly impacted my reviews as I could not understand your work (results) without this information. In any case, after reading other reviewers' comments and the response you provided to them, I am increasing my score.

---

> > > ### Author Response · Authors · 2022-08-08
> > > **Response to reviewer qHTA**
> > >
> > > Dear reviewer qHTA,
> > >
> > > Many thanks for improving your score. We will follow all reviewers' comments that provide very constructive and valuable improvements for our paper. If you have more suggestions, please kindly let us know. We are very glad to take them further.
> > >
> > > Best regards,
> > >
> > > Authors of Paper1270

---

### Official Review · Reviewer_eP7P · 2022-07-09

**Rating:** 8
**Confidence:** 3
**Soundness:** 4 excellent
**Presentation:** 4 excellent
**Contribution:** 4 excellent

**Summary:**

This paper presents a cross-dimensionality distillation approach based on the use of Hilbert curves. This facilitated the distillation of 3D network to 2D networks, with a variable-length method to encourage the 2D network to attend to activation features. Experiments were performed on two distinct datasets, compared to a set of existing knowledge distillation methods. Clear margins of improvement were observed in both datasets.

**Questions:**

A quick comment on the computation cost would be appreciated.

**Limitations:**

There did not seem to be discussion of the limitation of the presented work.

**Strengths And Weaknesses:**

The presented adoption of Hilbert curves to enable cross-dimensionality distillation is interesting, elegant, and well rationalized. The novelty is high.

The experimentation was thorough, and the baseline models are representative of existing approaches. The results show significant improvements over all baseline models in both datasets. The ablation analyses were also thorough.

I do not see notable weakness in the paper. One curiosity is regarding the computational cost of the construction of the Hilbert curve, and how does that affect the training of the model.

---

> ### Author Response · Authors · 2022-08-02
> **Response to reviewer eP7P**
>
> Thanks for your appreciation and constructive feedback! The computational cost from generating Hilbert Curve to finishing the Hilbert-based mapping function $\mathcal{H}_{n,p}$ is very low. We run the generation process on a single process on the Intel(R) Xeon(R) Silver 4216 CPU (2.10GHz) and calculate the computational costs on different sizes of feature maps. Results are presented as follows.
>
> | Time Consuming (ms)  |      |      |      |      |      |      |      |       |
> |---------------------|-------|-------|-------|-------|-------|-------|-------|--------|
> | $S$ (side length)      | 2     | 4     | 8     | 16    | 32    | 64    | 128   | 256    |
> | 2D (# points = $S^2$) | 0.034 | 0.046 | 0.090 | 0.204 | 0.458 | 2.505 | 2.319 | 5.166  |
> | 3D (# points = $S^3$) | 0.048 | 0.072 | 0.159 | 0.376 | 0.874 | 3.359 | 4.550 | 10.043 |
>
> What can be observed is that the mapping only costs 10ms even in processing a 3D feature map with the size 256 $\times$ 256 $\times$ 256. The time complexity of the generation process depends on the implementation of the Hilbert Curve. We adopt the most applied approach to generate Hilbert Curve in $O(\log{n})$ time. We agree that detailed discussion about the computational complexity of the generation process is necessary. We have added the discussion in the appendix of the revision. Please refer to Appendix "B. Computational Costs of Generating Hilbert Mapping Function" in the revised paper.

---

### Official Review · Reviewer_iQbj · 2022-07-09

**Rating:** 6
**Confidence:** 5
**Soundness:** 3 good
**Presentation:** 3 good
**Contribution:** 2 fair

**Summary:**

The paper studies 3D knowledge distillation for video analysis and medical imaging via Hilbert curve. The authors propose a Hilbert Distillation (HD), to explicitly distill structural information in intermediate features of 3D models. The authors further design Variable-length Hilbert Distillation (VHD) to dynamically shorten the walking stride of the Hilbert curve. The authors demonstrate the robustness of the method compared with some state-of-the-art methods.


**Questions:**

- Although the overall architecture is novel, its individual components are largely inspired by previous works [1 - 4]. In my humble view, the work seems to be very similar to the previous work. The technical novelty should be emphasized.
- Line # 169-170 “In reality, in the medical imaging task that 3D models are commonly applied, the spatial distribution of human organs are always fixed regardless of the data modality”. Why is the spatial distribution always fixed? Based on my understanding, the distribution of organs, especifically tumors,  should share a high variance [5 - 6, 12]. The authors should explore more medical examples instead of showing the special case.
- Line # 156 - 158 “only partial feature maps are activated and crucial for the final task”. Could you show some visualization to demonstrate these.
- As shown in Tables, the magnitude of the improvement of the proposed distillation-based methods remain unclear.
- I am wondering whether the authors can provide a more clear picture of how the Hibert distillation works. Although the results and the experiments look good, this is still the heart of the model, and it makes it hard for me to reason about the importance of the result and whether it is exhaustively defended? Do you have any theoretical proofs?
- Lack of comparing current state-of-the-art knowledge distillation based methods [5 - 8].
- More benchmarks in computer vision and medical imaging domains should be included to demonstrate the robustness of the proposed method.

Reference
[1] Self-Distillation Amplifies Regularization in Hilbert Space

[2] Overcoming the Curse of Dimensionality in Neural Networks

[3] Does knowledge distillation really work?

[4] Comparing kullback-leibler divergence and mean squared error loss in knowledge distillation

[5] Tumor‐infiltrating lymphocytes are a marker for microsatellite instability in colorectal carcinoma

[6] Integrative genomic profiling of large-cell neuroendocrine carcinomas reveals distinct subtypes of high-grade neuroendocrine lung tumors

[7] Contrastive representation distillation

[8] Heterogeneous knowledge distillation using information flow modeling

[9] Contrastive multiview coding

[10] Residual knowledge distillation

[11] Reskd: Residual-guided knowledge distillation

[12] Novel transfer learning approach for medical imaging with limited labeled data



**Limitations:**

See “Questions” above.


**Strengths And Weaknesses:**

##### Strengths
+ The paper proposes Hilbert Distillation (HD) to distill structural information from intermediate 3D feature maps.
+ The authors introduce Variable-length Hilbert Distillation (VHD) to efficiently transfer structural knowledge via Hibert curves.
+ The results are promising compared to some previous baselines.

##### Weaknesses
- The technical novelty of the method could be emphasized.
- The usefulness and motivation of the Hilbert curve is not clear.
- There are some claims that are not well supported with references, empirical or theoretical evidence (See “Question” - Q3)
- Results should include state-of-the-art methods from the literature.
- More benchmarks should be included to demonstrate the robustness of the proposed method.

---

> ### Author Response · Authors · 2022-08-02
> **Response to reviewer iQbj -- part 4**
>
> >Q7: More benchmarks in computer vision and medical imaging domains should be included to demonstrate the robustness of the proposed method.
>
> A7: Thanks for your advice. We have continued some extra experiments on Kinetics-400 after the submission. Since they are not included in the original submission, we listed the table in the following:
>
> | Kinetics-400   |                 |                 |
> |----------------|------------------|------------------|
> | Teacher        | 3DResNet-50      | 3DResNet-50      |
> | Student        | ResNet-50        | VGG16            |
> | Teacher        | 74.15            | 74.15            |
> | Student        | 67.20 $\pm$ 0.23 | 65.43 $\pm$ 0.19 |
> | KD             | 68.03 $\pm$ 0.31 | 66.71 $\pm$ 0.46 |
> | SP             | 69.14 $\pm$ 0.48 | 68.29 $\pm$ 0.55 |
> | PKT            | 68.37 $\pm$ 0.29 | 67.35 $\pm$ 0.41 |
> | AT (with avg)  | 68.21 $\pm$ 0.59 | 66.86 $\pm$ 0.30 |
> | AT (with cnv)  | 68.35 $\pm$ 0.44 | 67.14 $\pm$ 0.47 |
> | AT (with max)  | 68.13 $\pm$ 0.32 | 67.19 $\pm$ 0.44 |
> | AFD (with avg) | 68.82 $\pm$ 0.65 | 67.77 $\pm$ 0.58 |
> | AFD (with cnv) | 69.59 $\pm$ 1.02 | 67.90 $\pm$ 0.81 |
> | AFD (with max) | 69.08 $\pm$ 0.64 | 68.48 $\pm$ 0.59 |
> | HD (ours)      | 70.28 $\pm$ 0.36 | 69.82 $\pm$ 0.42 |
> | VHD (ours)     | 70.91 $\pm$ 0.85 | 70.40 $\pm$ 0.73 |
>
> The results demonstrated that our method still outperforms the best performance holder AFD and SP in the current completed experiments. We have added the results in the appendix of the revision. Please refer to Appendix "D. More Benchmarks" in the revised paper. We will also present the experiments on Kinetics-400 of the same scale as the existing ActivityNets experiment in the final version.
>
> Reference
>
> [r1] Self-Distillation Amplifies Regularization in Hilbert Space
>
> [r2] Overcoming the Curse of Dimensionality in Neural Networks
>
> [r3] Does knowledge distillation really work?
>
> [r4] Comparing kullback-leibler divergence and mean squared error loss in knowledge distillation
>
> [r5] Tumor‐infiltrating lymphocytes are a marker for microsatellite instability in colorectal carcinoma
>
> [r6] Integrative genomic profiling of large-cell neuroendocrine carcinomas reveals distinct subtypes of high-grade neuroendocrine lung tumors
>
> [r7] Contrastive representation distillation
>
> [r8] Heterogeneous knowledge distillation using information flow modeling
>
> [r9] Contrastive multiview coding
>
> [r10] Residual knowledge distillation
>
> [r11] Reskd: Residual-guided knowledge distillation
>
> [r12] Novel transfer learning approach for medical imaging with limited labeled data
>
> [1] Levitan, B.M. (2001), "Hilbert space", Encyclopedia of Mathematics, EMS Press.
>
> [2] Über die stetige Abbildung einer Linie auf ein Flächenstück
>
> [3] Sur une courbe, qui remplit toute une aire plane
>
> [4] Grad-cam: Visual explanations from deep networks via gradient-based localization
>
> [5] Learning Deep Features for Discriminative Localization
>
> [6] Distilling Holistic Knowledge with Graph Neural Networks
>
> [7] Analysis of the Clustering Properties of the Hilbert Space-Filing Curve
>
> [8] Analysis of the Hilbert curve for representing two-dimensional space
>
> [9] On Multidimensional Curves with Hilbert Property

---

> > ### Comment · Reviewer_iQbj · 2022-08-09
> > **Thank you for your clarifications!**
> >
> > I checked the reviews and the author feedbacks, as well the updated manuscript. I feel the manuscript is improved from the firstly submitted one. With this understanding, I am increasing my score to 6.
> >
> > Some of the ideas here may be familiar to readers of the following papers [1, 2]. In my humble opinion, if these studies have any relevance to the topic at hand, it would be great if the authors would highlight them in the Related Work.
> >
> > Reference:
> >
> > [1] Bootstrapping Semi-supervised Medical Image Segmentation with Anatomical-aware Contrastive Distillation
> >
> > [2] SimCVD: Simple contrastive voxel-wise representation distillation for semi-supervised medical image segmentation
> >
> > [3] Self-supervised Contrastive Cross-Modality Representation Learning for Spoken Question Answering
> >
> > [4] Momentum contrastive voxel-wise representation learning for semi-supervised volumetric medical image segmentation
> >
> > [5] Understanding Dimensional Collapse in Contrastive Self-supervised Learning

---

> > > ### Author Response · Authors · 2022-08-09
> > > **Many thanks for improving your score!**
> > >
> > > Dear reviewer iQbj,
> > >
> > > We truly appreciate for recommending the interesting works[1, 2]. In the revision, we will discuss their contribution and highlight the difference with our work in the Related Works. Moreover, we will follow all your constructive comments to improve our paper. Please kindly let us know if you have more suggestions. We are very glad to take them further.
> > >
> > > Best regards,
> > >
> > > Authors of Paper1270

---

> ### Author Response · Authors · 2022-08-02
> **Response to reviewer iQbj -- part 3**
>
> >Q5: I am wondering whether the authors can provide a more clear picture of how the Hilbert distillation works. Although the results and the experiments look good, this is still the heart of the model, and it makes it hard for me to reason about the importance of the result and whether it is exhaustively defended? Do you have any theoretical proofs?
>
> A5: Thanks for the great question. We have provided two new figures in the appendix in the revision to depict how the method works. Please check Figs. 7 and 8 in Appendix "A. More Details About How Hilbert Distillation Works" in the revised paper. Fig. 7 shows the case of converting a 16 $\times$ 16 2D space with two semantic pixel-level classes to 1D space using the Hilbert Curve. The pixels of class 2 indicated by the yellow area are still largely aggregated after finishing the mapping by walking along with the Hilbert Curve, which means the structural information is well preserved in the dimension reduction. Fig. 8 presents how the proposed Hilbert Distillation works from the perspective of activation features. As the illustrated 3D-to-2D distillation example on COVID classification, the Hilbert Curve can help aggregate lung features into the 1-dimensional space and preserve the feature continuity. Then the student features in the 2D lung area can directly learn from teacher features in the 3D lung area.
>
> The core ability of Hilbert Distillation is from the trait that the Hilbert Curve can reduce the dimension as well as preserve the locality of data points (features). Benefiting from the curve's mapping rule, two data points which are close to each other in the space-filling curve are also close to each other in the original space. Some previous works (e.g., [7]) have analyzed this kind of clustering property. The Hilbert Curve is founded on artificial rules and is difficult to derive with a short piece of mathematical formulas though it has been widely applied in some specific areas in recent years. It is recommended to refer to [8] (2D dimension) and [9] (multi-dimension) if you need more information about the analysis of the Hilbert Curve. We would emphasize that we do not alter the theory of the Hilbert Curve. Our method consists of 1) skillfully adopting the Hilbert Curve to conduct distillation between cross-dimensionality networks; 2) dynamically changing the walking stride of the Hilbert Curve in activation areas to improve the distillation performance further.
>
> >Q6: Lack of comparing current state-of-the-art knowledge distillation based methods [r5 - r8].
>
> A6: Thanks for recommending the related SOTA works. If we understand right, the SOTA methods you recommended are actually [r7] - [r11]. We have added the experiments of [r8], [r10], and [r11] on ActivityNet and Large-COVID-19. Please check the table presented below. As most recent knowledge distillation based methods, the added methods only consider the 2D-to-2D distillation thereby they need the help of alignment functions as what we have analyzed in Line # 261 - 266. Our method outperforms previous approaches on 3D-to-2D distillation because we largely focus on preserving structural information of 3D feature maps for 2D student networks to learn. We truly appreciate for recommending the contrastive learning based methods [r7, r9]. The reason why they are not included in the experiment is because the student requires the same/augmented samples of the teacher to construct "positive pair", which is impossible in cross-dimensionality distillation as the input dimension is different. We agree that detailed discussion about the adaptability of contrastive learning based methods on 3D-to-2D distillation problems in our paper is necessary. We have added the discussion in the appendix of the revision. Please refer to Appendix "C. The Adaptability of 2D-to-2D Distillation Methods on 3D-to-2D Distillation Problems" in the revised paper.
>
> |                      | ActivityNet      | Large-COVID-19   |
> |-----------------------|------------------|------------------|
> | HKD[r8] (with avg)    | 61.85 $\pm$ 0.26 | 82.03 $\pm$ 0.45 |
> | HKD[r8] (with cnv)    | 62.20 $\pm$ 0.47 | 81.84 $\pm$ 0.29 |
> | HKD[r8] (with max)    | 62.13 $\pm$ 0.19 | 82.31 $\pm$ 0.41 |
> | RKD[r10] (with avg)   | 62.37 $\pm$ 0.33 | 82.50 $\pm$ 0.37 |
> | RKD[r10] (with cnv)   | 62.42 $\pm$ 0.51 | 82.73 $\pm$ 0.56 |
> | RKD[r10] (with max)   | 61.97 $\pm$ 0.48 | 82.76 $\pm$ 0.39 |
> | ResKD[r11] (with avg) | 62.74 $\pm$ 0.62 | 82.61 $\pm$ 0.82 |
> | ResKD[r11] (with cnv) | 62.89 $\pm$ 0.76 | 83.02 $\pm$ 0.78 |
> | ResKD[r11] (with max) | 62.74 $\pm$ 0.62 | 83.27 $\pm$ 0.71 |
> | HD (ours)                    | 63.55 $\pm$ 0.28 | 85.05 $\pm$ 0.58 |
> | VHD (ours)                  | 63.71 $\pm$ 0.63 | 85.55 $\pm$ 0.72 |
>
> Note that this is a one-off table for this response. The results will be integrated into Fig.3 of the main body in the revised paper to keep the consistent style for comparable methods.

---

> ### Author Response · Authors · 2022-08-02
> **Response to reviewer iQbj -- part 2**
>
> >Q2: Line # 169-170 “In reality, in the medical imaging task that 3D models are commonly applied, the spatial distribution of human organs are always fixed regardless of the data modality”. Why is the spatial distribution always fixed? Based on my understanding, the distribution of organs, especifically tumors, should share a high variance [r5 - r6, r12]. The authors should explore more medical examples instead of showing the special case.
>
> A2: Thanks for your constructive comment! We agree with you that the original description only represents partial medical examples. In this paper, the subsection 3.2.a (line #166-173) has clarified the adaptability of the proposed method. First, the method can naturally handle structurally invariable cases such as stable organ recognition in computerized tomography (CT) because the relative positions of the activation features are well aligned between the converted student and teacher 1D space. Moreover, the proposed method is able to deal with activation feature alignment problems in high variance distribution cases (video recognition) by appending an extra simple length-preserving fully connection layer. As stated in your comment that some medical examples also share the high variance distribution, we agree that more detailed discussion of this question in the subsection is necessary. Thus, we will add the discussion in the main body of the revised paper. Nevertheless, it is worth noting that this would not affect the adaptability and novelty of our method because our method will deal with these medical examples as the same as video recognition (please refer to our experiments for the video dataset).
>
> >Q3: Line # 156 - 158 “only partial feature maps are activated and crucial for the final task”. Could you show some visualization to demonstrate these.
>
> A3: Thanks for your advice. We have added some visualization in the appendix of the revision. Please check Fig. 6 in Appendix "A. More Details About How Hilbert Distillation Works" in the revised paper. The question you raised can be  supported by the existing research about interpretability for convolutional neural networks [4, 5], which can find out the important (activation) features in feature maps. In addition, many previous works about interpretability and Attention mechanism demonstrate that only a fraction of a neural network plays a crucial role. The visualization in [5] can also give the intuition to understand the argument.
>
> >Q4: As shown in Tables, the magnitude of the improvement of the proposed distillation-based methods remain unclear.
>
> A4: Thanks for your advice. Following the the previous works [r7, 6], we have added the Average Relative Improvement (ARI) in the table 1 as listed in the following:
>
> |  |     ActivityNet        |             |  |     Large-COVID-19        |             |         |
> |-------------|-------------|-------------|----------------|-------------|-------------|---------|
> |             | 3DResNet-50 | 3DResNet-50 | ARI (%)        | 3DResNet-50 | 3DResNet-50 | ARI (%) |   |
> |             | ResNet-50   | VGG16       |                | ResNet-50   | VGG16       |         |   |
> | Teacher     | 71.34       | 71.34       | /              | 90.15       | 90.15       | /       |   |
> | Student     | 61.42       | 60.22       | /              | 79.92       | 77.4        | /       |   |
> | KD          | 62.30       | 61.45       | 184.59%        | 82.08       | 82.37       | 110.51% |   |
> | SP          | 62.88       | 62.27       | 71.11%         | 83.69       | 82.85       | 47.79%  |   |
> | PKT         | 62.73       | 62.70       | 64.02%         | 83.20       | 82.69       | 61.15%  |   |
> | RKD         | 62.14       | 61.23       | 247.15%        | 83.02       | 82.44       | 69.87%  |   |
> | CCKD        | 62.78       | 61.88       | 98.65%         | 83.19       | 82.70       | 61.27%  |   |
> | HD (ours)         | 63.55       | 63.46       | 12.40%         | 85.05       | 84.63       | 9.99%   |   |
> | VHD (ours)         | 63.71       | 64.02       | /              | 85.55       | 85.37       | /       |
>
> The value of ARI can be calculated as $ARI=\frac{1}{M} \sum_{i=1}^{M} \frac{Acc_{VHD}^{i}-Acc_{BKD}^{i}}{Acc_{BKD}^{i}-Acc_{STU}^{i}} \times 100 \%$, where M is the number of different architecture combinations, and BKD and STU refer to the baseline methods and student network, respectively. In brief, ARI presents the magnitude of the improvement of the proposed VHD compared with the method located in that row. We will load this revision in the main body of the paper.

---

> ### Author Response · Authors · 2022-08-02
> **Response to reviewer iQbj -- part 1**
>
> Many thanks for your valuable comments and constructive feedback! We will answer your concerns in the following.
>
> >Q1: Although the overall architecture is novel, its individual components are largely inspired by previous works [r1 - r4]. In my humble view, the work seems to be very similar to the previous work. The technical novelty should be emphasized.
>
> A1: Thank you for identifying the novelty of our architecture. In fact, Hilbert Space[1] and Hilbert Curve[2] is different in terms of concept and applications. A Hilbert Space is a real or complex inner product space, which is generalized by Euclidean space. The Hilbert Space belongs to the space theory and is commonly applied to mathematics and physics fields. The Hilbert Curve is a continuous fractal space-filling curve that gives a mapping between 1D and 2D (or higher) space, as a variant of the space-filling Peano curves[3]. The Hilbert Curve is essentially a dimensionality reduction method based on a space walk strategy, which is commonly applied in scheduling and signal processing. Therefore, Hilbert Curve and Hilbert Space differ from concept and application perspectives.
>
> The works [r1, r2] mentioned by the reviewer focus on studying the feasibility and interpretability of knowledge distillation (KD) in the Hilbert Space, which are totally different from our method that utilizes the Hilbert Curve to cope with cross-dimensionality distillation. The works [r3, r4] study the interpretability of existing KD methods, which differ from our method that proposes a new distillation method for a specific scenario, 3D-to-2D distillation.
>
> We argue that our individual components are novel. Compared with the previous works, the main technical contributions of this work are summarized as follows.
>
> 1. We propose to adopt the construction process of the Hilbert Curve to help transfer dimensionally heterogeneous feature maps to an informative and distillable 1D representation effectively. To the best of our knowledge, this is the first work that facilitates the space-walk strategy (e.g., the Hilbert Curve) for efficiently Cross-Dimensionality distillation in the field. The reason why the method works, please refer the answer A5 of your question Q5.
> 2. We further propose the Variable-length Hilbert Distillation to make the Hilbert Distillation pay more attention to distilling activation (a.k.a. important) features. The activation features are found by calculating gradients for all the objective classes. About the definition of activation features, please refer to the answer A3 of your question Q3.
> 3. Although the cross-dimensionality distillation is of importance, the efficient solutions for the cross-dimensionality distillation problem have been rarely studied. The related research and discussions that can be referred to are very limited. This paper presents a superior approach for cross-dimensionality distillation by leveraging the theory of Hibert Curve. We believe our effective and reproductive approach can take the lead in coping with the significant and practical distillation problem.

---

### Meta-Review · Area_Chair_fjz2 · 2022-08-26

**Recommendation:** Accept
**Confidence:** Certain

**Metareview:**

This submission was reviewed by four reviewers. All reviewers provided detailed and informative reviews. During rebuttal, the authors actively submitted detailed rebuttals, which lead to improved evaluations by the reviewers with improved scores. Overall, this is an interesting paper and an accept is recommended.

**Award:**

No

---

### Decision · Program_Chairs · 2022-09-14

Accept